# Research of Liquid-Crystal Materials for a High-Performance FFS-TFT Display

**DOI:** 10.3390/molecules28020754

**Published:** 2023-01-11

**Authors:** Haiguang Chen, Youran Liu, Maoxian Chen, Tianmeng Jiang, Lanying Zhang, Zhou Yang, Huai Yang

**Affiliations:** 1School of Materials Science and Engineering, University of Science and Technology Beijing, Beijing 100083, China; 2Beijing Bayi Space Liquid Crystal Material Technology Company, Yangfang, Beijing 102200, China; 3Beijing Key Laboratory of Liquid Crystal Materials Analysis and Application Technology, Beijing 102502, China; 4School of Materials Science and Engineering, Peking University, Beijing 100871, China

**Keywords:** liquid crystal, FFS-TFT display, high performance

## Abstract

A novel liquid-crystal compound of more than 99.95% purity with high performance (such as a high clearing point, large dielectric anisotropy, high optical anisotropy, low viscosity, and large elastic constants) was designed and synthesized according to the fringe-field switching thin-film-transistor-liquid-crystal display requirements (FFS-TFT). Then, a mixed liquid-crystal material suitable for an FFS-TFT display was developed by mixing this compound with other reported compounds, developing a product whose quality was that of the highest level of similar foreign products and which fully met the customer’s use requirements (BOE), and thus able to completely replace similar imported materials.

## 1. Introduction

In 1888, the Austrian botanist F.Reinitzer [1] first observed the liquid-crystal phenomenon, and the following year, physicist O. Lehmann [2] named it “Liquid Crystals”. In the late 1960s, Heilmeier [3], from the RCA Company in the United States, used liquid crystal for the “Dynamic Scattering Effect”, which was the first application of a liquid crystal in the field of display. Now fringe-field switching (FFS-LCD) is a favorable choice for panels because of its high performance. This TFT-LCD, as one of the most advanced LCD modes, also solves the shortcomings of conventional LCDs caused by its principle of operation and the characteristics of liquid-crystal materials [4,5,6,7,8,9] such as (1) a small display angle, (2) a slow response speed, (3) a narrow working temperature range, and (4) the requirement of backlighting when there is no external light. However, there have also been high technical performance requirements for liquid-crystal materials [10,11,12,13,14,15,16,17], which are listed in Table 1.

At present, high-performance mixed liquid-crystal materials are almost monopolized by foreign suppliers, and there is an urgent requirement to change this situation. Therefore, in this paper, a high-performance mixed liquid-crystal material suitable for TFT-LCD displays was developed in order to replace imported products.

## 2. Results and Discussions 

### 2.1. Technical Route

Appendix A shows the technical route for the design of mixed liquid-crystal materials with high performance. Firstly, liquid-crystal compounds with high performance are designed based on the relationships between the molecular structures and the physical properties of liquid-crystal materials. Secondly, the mixed liquid-crystal materials with high performance, which meet the requirements of FFS-TFT, were prepared by combining some known compounds of required performance.

### 2.2. Technical Specification Requirements

The research is based on the request from BOE Optoelectronics Technology Co. Ltd., (Beijing, China), a well-known domestic panel manufacturer, which requires the development of a domestic material to replace the LCD material produced by Japan JNC (Japan’s new CHISSO) for the TFT-FFS screens of 5-inch mobile phones, and its performance parameters must meet the specifications shown in Table 2. In order to meet the reliability requirements of the liquid-crystal display, the materials must also meet the quality and reliability requirements shown in Appendix A.

### 2.3. Design and Performance Evaluation of High-Performance Liquid-Crystal Compounds

#### 2.3.1. Structure Determination of the High-Performance Liquid-Crystal Compounds 

According to the demand for high performance, it is necessary to design a new liquid-crystal compound with a big Δ*n* and Δ*ε*, a high clearing point, and a small γ1. From the relationship between structure and performance as shown in Appendix A, the conventional liquid-crystal compound shown in Figure 1 is commonly used [18].

To develop materials with properties close to Figure 1, an additional F atom was designed in the alkyl chain on the left side, as shown in Figure 2, to create a structure that has not been reported in the literature so far. 

By simulating the dipole moments of Figure 1 and Figure 2 with molecular simulation software, it was found that the performance is similar. Then, in this paper, the compound was synthesized.

#### 2.3.2. Synthesis of High-Performance Liquid-Crystal Compounds 

According to the literature report [19], a crude single crystal with a purity of 98.857% was synthesized, as shown in Appendix A. A single crystal with a purity of 99.96%, which fully met the standard of TFT liquid-crystal compounds, was produced using recrystallization and column chromatography, 

#### 2.3.3. Performance Evaluation of the High-Performance Liquid-Crystal Compound 

The physical properties of the synthesized compound were tested, and it can be seen from the results shown in Table 3, that the compound has obvious advantages over the traditional structure.

### 2.4. The Preparation of the Liquid-Crystal Material Mixture

#### 2.4.1. The Method of Allocating Mixed Liquid Crystal

The allocating of mixed liquid crystal is to mix certain proportions of compounds that can improve some properties of the liquid crystal to meet the requirements of different display modes, [20] for there is no single compound that can meet the requirements of a liquid crystal display. The most important criterion for modulating mixed liquid crystal is to master the relationship between the device’s performance and the liquid crystal’s physical performance, as described in Section 1. Based on this, the newly synthesized compound was used to adjust the properties of mixed liquid-crystals, such as the Δ*ε* and *K* value, etc.

#### 2.4.2. Debugging Rules and Test Methods for Mixed Liquid Crystal—Adjustment and Test Method for Clear Point (TNI)

The temperature range of the liquid-crystal phase is an important physical parameter to ensure the use is within the temperature range of the display, and the clearing point of the liquid crystal directly determines the upper limit of the working temperature. For mixed liquid crystal, the value of the clearing point is shown in Equation (1).
(1)$$T_{NI}=\frac{X_{1}\Delta H_{1}+X_{2}\Delta H_{2}}{X_{1}\Delta S_{1}+X_{2}\Delta S_{2}}$$

*X*_1_ and *X*_2_ refer to grams per percent of components 1 and 2.

The introduction of tetracyclic compounds, as shown in Equation (2), can improve the clearing point of the new compounds. In this paper, the Q10 DSC of the American TA Company was used to measure the clear point of mixed liquid crystal.

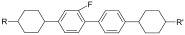
(2)

#### 2.4.3. Adjustment and Test Method of Δ*n*

For mixed crystals, the value of Δ*n* is shown in Equation (3) [21].
(3)$$\Delta n=\frac{\left(X_{1}/N_{1}\right)\Delta n_{1}+\left(X_{2}/N_{2}\right)\Delta n_{2}}{X_{1}/N_{1}+X_{2}/N_{2}}$$

*N*_1_ and *N*_2_ refer to the density of components 1 and 2.

Δ*n* can be increased by introducing π electrons such as the triphenyl ring shown in Equation (4); on the contrary, it can be reduced by introducing cyclohexane. Δ*n* is completed using a Japanese AITO DR-M2 Abbe refractometer.

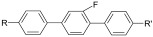
(4)

#### 2.4.4. Adjustment and Test Method of Δ*ε*

For mixed crystal, the value of Δ*ε* is shown in Equation (5).
(5)$$\Delta \varepsilon =\frac{\left(X_{1}/N_{1}\right)\Delta \varepsilon_{1}+\left(X_{2}/N_{2}\right)\Delta \varepsilon_{2}}{X_{1}/N_{1}+X_{2}/N_{2}}$$

The Δ*ε* is generally improved by introducing link groups and adding polar terminal groups. The Δ*ε* is completed using the E4980A LCR instrument.

#### 2.4.5. Adjustment and Test Method of Low-Temperature Miscibility

The evaluation of low-temperature miscibility includes several methods and means, which need to be used together.

Observation method I

Place the prepared mixed liquid crystal into a sample bottle, which is placed in a low-temperature observation box (the temperature is generally controlled at −20 °C or −30 °C), and observed every day for 10 days to see whether there is crystal precipitation. It is determined as OK only if there is no crystal precipitation.

2.Observation method II

The test cell (3.5 μm, FFS) with mixed liquid crystal is placed in the low-temperature observation box (the temperature is generally controlled at −30 °C or −40 °C) and observed every day for 10 days to see whether there is crystal precipitation. It is determined as OK only if there is no crystal precipitation.

3.DSC spectrum

The DSC test method is as follows: lower the temperature to −85 °C, then program the temperature to observe whether there is an obvious crystallization endothermic peak and whether the glass transition temperature is lower than −70 °C. Combined with the above three methods, the mixed liquid crystal is considered OK only if there is no abnormality.

#### 2.4.6. Adjustment and Test Method of *K*

For mixed crystals, the value of *K* is shown in Equation (6) [22].
(6)$$\sqrt{K_{ii}}=\frac{X_{1}\sqrt{K_{ii1}}+X_{2}\sqrt{K_{ii2}}}{X_{1}+X_{2}}\left(i=1,2,3\right)$$
where *X*_1_ and *X*_2_ are the molar numbers of components, and *K_ii_*_1_ and *K_ii_*_2_ are the elastic constants of each component. As mentioned above, the *K* value of the long-chain monomer, especially the four-ring monomer, is relatively large, and the new structural monomer Equation (7) has a large *K* value.

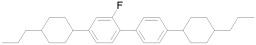
(7)

In this paper, the comprehensive ALCT-PP1 liquid-crystal performance tester was used to measure the elastic constants of mixed liquid crystals.

#### 2.4.7. Adjustment and Test Method of γ1

In order to reduce γ1, it needs more components with a low γ1 [23,24], as shown in Equation (8):
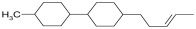
(8)

## 3. Materials and Methods 

### 3.1. Experimental Result

After systematic experimental research, a high-performance mixed liquid crystal (commercially named BHR95300) was obtained, as shown in Table 4. The physical and quality properties of BHR95300 were tested. The results are shown in Table 5 and Table 6.

### 3.2. Physical Performance Test

It can be seen from Table 5 that the physical performance parameters of the mixed liquid crystal BHR95300 are basically consistent with the foreign product ZBE5311, with a slightly slower response time, but overall, it can meet the specification requirements.

It can be seen from Table 6 that the *K* value (K11, K22, and K33) of the mixed liquid crystal BHR95300 is basically the same as ZBE5311, with 20% higher γ1, which is the main cause of the slower response time.

### 3.3. Quality Performance Test

It can be seen from Table 7 that the quality and performance indexes of the mixed liquid crystal material can fully meet the development requirements of the customer, and some indexes are even better than similar foreign products.

It can be seen from Appendix A that the content of metal ions fully meets the specification requirements of less than 20PPb.

The mixed liquid crystal materials that we developed were provided to BOE Optoelectronics Technology Co., Ltd. (Beijing, China) for localization. After BOE’s evaluation, the performance indexes such as the chromaticity, gamma voltage, contrast, and response time of the mixed liquid crystal, as well as the residual image, Flicker, and high and low-temperature reliability completely passed the customer’s performance evaluation. The product can completely replace imported materials.

## 4. Conclusions

Fringe-field switching (FFS-LCD) is a favorable choice for panels because of its high performance [25,26]. However, there are high technical performance requirements for the liquid-crystal materials of FFS-LCDs [27,28]. In this paper, a liquid-crystal compound with a stable structure and high performance was designed and synthesized. Through fine purification operations, its purity index reached more than 99.95%, which fully met the requirement for the liquid-crystal materials used in a TFT-LCD display. A mixed liquid-crystal material BHR95300 was developed by combining the synthesized liquid-crystal compound above with other reported compounds, yielding better characteristic properties than the imported materials, such as a 2.2 × 10^14^ specific resistivity, a 99.4% VHR, a 24.3 ion density, and the advantages were even more obvious after UV irradiation. The performance of the mixed liquid-crystal material BHR95300 meets the demands of a high-quality FFS TFT-LCD and has reached the quality level of the most advanced products at this time.

## Figures and Tables

**Figure 1 molecules-28-00754-f001:**
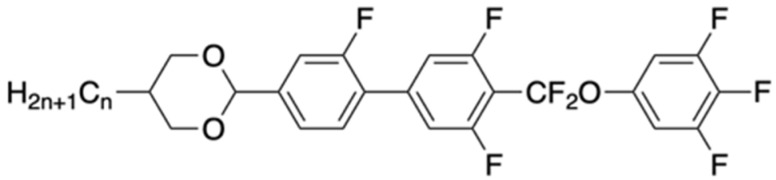
The conventional liquid-crystal compound.

**Figure 2 molecules-28-00754-f002:**
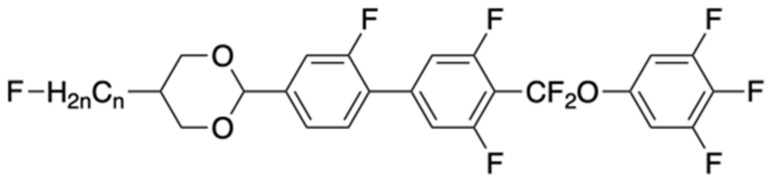
The designed structure of liquid-crystal compound.

**Table 1 molecules-28-00754-t001:** Performance requirements of a fringe-field switching-thin-film-transistor liquid-crystal display (FFS TFT-LCD).

Entry	Device Performance	Requirements for LCs
1	A low driving voltage (energy saving)	A low Vth
2	Fast response (suitable for dynamic display)	Low rotational viscosity (γ1), a large *K* value
3	High transmittance (energy saving)	Small Δ*ε*, a large off-state dielectric constant
4	A high contrast (improves the display effect)	A large *K* value, small *n_o_*, *n_e_*, and (Δ*n*)
5	Suitable Δnd value (consistent with the requirements)	A cell thickness (d) necessary for matching with Δ*n*
6	A wide temperature range	Large Cp (≥80 °C), Δ*n*, Δ*ε*, and *K*
7	High reliability (long service life)	High VHR (≥97%), low ion, high resistivity and purity, and high UV and thermal stability

**Table 2 molecules-28-00754-t002:** Specifications of the physical parameters.

Liquid-Crystal Property		Specification
Clear point	TNI [°C]	≥90
Low-temperature storage	TSN [°C]	≤−30
Dielectric anisotropy [25 °C, 1.0 kHz]	Δ*ε*	5.6 ± 0.2
Optical anisotropy [589.3 nm, 25 °C]	Δ*n*	0.100 ± 0.002
Elastic constant [25 °C]	K11 [pN]	13.4 ± 0.5
	K22 [pN]	6.7 ± 0.5
	K33 [pN]	16.7 ± 0.5
γ1 [25 °C]	[mpa.s]	≤80
Response Time [25 °C]	R.T [ms]	≤30

**Table 3 molecules-28-00754-t003:** The contrast in performance.

Physical Property Parameters of Liquid Crystal								
Structure of the Compounds	Δ*ε* [25 °C]	Δ*n* [25 °C]	*T_S-N_* [°C]	Cp [°C]	γ1 [mPa.s]	K11 [pN]	K22 [pN]	K33 [pN]
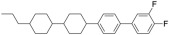	12.9	0.1734	102.9	258.8	773.1	25.4	12.7	36.0
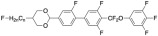	34.5	0.268	82.9	102	225.6	27.5	14.0	38.0

**Table 4 molecules-28-00754-t004:** The constitute of formula.

Entry	Structural	Percentage	Remarks
1	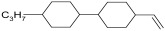	36%	Reduce γ1
2	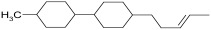	10%	Reduce γ1
3	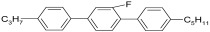	5%	Increase Δ*n*
4	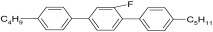	5%	Increase Δ*n*
5	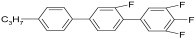	5%	Increase Δ*ε*
6	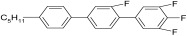	5%	Increase Δ*ε*
7	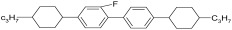	3%	Increase Cp, *K*
8	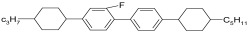	3%	Increase Cp, *K*
9	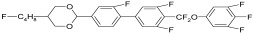	13%	Increase Cp, Δ*n*, *K*
10	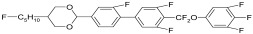	15%	Increase Cp, Δ*n*, *K*

**Table 5 molecules-28-00754-t005:** Physical Parameters.

Liquid-Crystal Property		Spec	ZBE-5311	BHR95300	Result
Clear point	T_NI_ [°C]	≥90	90.6	93.0	OK
Low-temperature Storage	T_SN_ [°C]	≤−30	≤−30	≤−30	OK
Dielectric anisotropy [25 °C, 1.0 kHz]	Δ*ε*	+5.6 ± 0.2	+5.6	+5.6	OK
	*ε* _‖_		8.5	8.5	OK
	*ε* _⊥_		2.9	2.9	OK
Optical anisotropy [589.3 nm, 25 °C]	Δ*n*	0.100 ± 0.002	0.101	0.100	OK
	n_e_		1.585	1.573	OK
	n_o_		1.484	1.473	OK
Response Time [25 °C]	R.T [ms]	≤30	21.50	25.7	OK

**Table 6 molecules-28-00754-t006:** Elastin Constant and γ1.

Liquid-Crystal Property		Spec	ZBE-5311	BHR95300	Result
Elastic constant [25 °C]	K_11_ [pN]	13.4 ± 0.5	13.4	13.2	OK
	K_22_ [pN]	6.7 ± 0.5	6.7	6.6	OK
	K_33_ [pN]	16.7 ± 0.5	16.7	17.2	OK
Rotary Viscosity	γ1 [25 °C, mpa.s]	≤80	63	78	OK

**Table 7 molecules-28-00754-t007:** Quality Parameters.

Liquid-Crystal Property		Spec	ZBE5311	BHR95300	Result
Specific resistivity [25 °C]	ρ [Ω·cm]	≥1 × 10^13^	1.8 × 10^14^	2.2 × 10^14^	OK
Specific resistivity [25 °C] After heat 120 °C, 2 h	ρ [Ω·cm]	≥1 × 10^12^	5.4 × 10^13^	6.2 × 10^13^	OK
resistivity [25 °C] After UV 10,000 mW/cm^2^	ρ [Ω·cm]	≥1 × 10^12^	8.9 × 10^13^	9.1 × 10^13^	OK
VHR [1 V, 2 s, 60 °C]	(%)	≥98.5%	99.3%	99.4%	OK
VHR [1 V, 2 s, 60 °C] After UV 10,000 mW/cm^2^	(%)	≥98%	99.1%	99.1%	OK
Ion density [1 V, 0.01 Hz, 60 °C]	PC	≤50	25.6	24.3	OK
Ion density [1 V, 0.01 Hz, 60 °C] After UV 10,000 mW/cm^2^	PC	≤100	76.5	54.2	OK

## Data Availability

Samples of the compounds are available from the authors.

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
