# Peer review of "Research of Liquid-Crystal Materials for a High-Performance FFS-TFT Display"

_molecules, 2023, doi:10.3390/molecules28020754_

Round 1
Reviewer 1 Report
General comments
The article is quite interesting; however, there are some parts that need to be improved. There are mixed parts, in my opinion you should be restructured. The discussion part is a bit poor. It should be improved by including more international literature.
Specific comments:
Materials and Methods and Results:
Why present these three parts together?
Material and methods must be adequately explained.
Results and Discussions:
Line 82: 2.41.Experimental Method. Why does this point appear in this section?
Discussion
The discussion part is a bit poor. It should be improved by including more international literature. This section should not repeat results.
Author Response
Referee 1:
General Comments
The article is quite interesting; however, there are some parts that need to be improved. There are mixed parts, in my opinion you should be restructured. The discussion part is a bit poor. It should be improved by including more international literature.
Specific Comments
Materials and Methods and Results:
Why present these three parts together?
Material and methods must be adequately explained.
Answers:
We modified Materials & Methods & Results into Materials & Methods, and classified Mixed Liquid Crystal Formula into Materials & Methods for detailed description.
Results and Discussions:
Line 82: 2.41.Experimental Method. Why does this point appear in this section?
Answers:
We modified “Experimental Method” into “The Method of allocating mixed liquid crystal”, it’s to explain the method of adjust the properties of mixed liquid crystals, such as Δε and K value, etc.
Discussion
The discussion part is a bit poor. It should be improved by including more international literature. This section should not repeat results.
Answers:
We have increased the content of discussion part and introduced international literatures 27-30.
Reviewer 2 Report
The paper reports a new material for fringe-field switching LCD application. The compound is based on a structure known from the prior art, however the fluorine-substituted hydrogen in the alkyl tail allows for considering the material as an original one. The paper is well structured, the new material as well as LC mixture based on it is thoroughly studied and comprehensively described. The developed LC mixture meets the high standards of the industry. The paper will be useful for those dealing with synthesis and formulation of LC materials. I recommend to publish it, provided that the authors address the following remarks:
· Line 26. “heilmeiert” should be corrected to “Heilmeier”. The corresponding reference [3] is probably incorrect, should be changed to a relevant paper by Heilmeier, such as “Heilmeier GH, Zanoni LA. Electro-optical device” US patent 3,499,112 A, 1967” or “Heilmeier GH, Zanoni LA, Barton LA. Dynamic scattering: a new electrooptic effect in certain classes of nematic liquid crystals. Proc IEEE. 1968;56:1162–1171”
· The comparison of LCD to Cathode Ray Tube seems a bit out-of-date, I suppose that attention to competing of FFS with IPS and other LCD modes instead would better match with the title and would be beneficial for the introduction.
· The values Cp and VHR should be denoted
· Table 3 should contain units, when applicable
Minor errors and typos:
· Lines 27-37. References [4-9] are missing in the text
· Line 91. Typo
· X1, X2, N1, N2 should be denoted when first mentioned
· Formulas (5) and (7) should be changed places
· Line 102 – unclear language: “… it can be reduced by introducing cyclohexane such as formula 6”. Formula 6 does not contain cyclohexane.
Author Response
Referee 2:
Comments to the author
The paper reports a new material for fringe-field switching LCD application. The compound is based on a structure known from the prior art, however the fluorine-substituted hydrogen in the alkyl tail allows for considering the material as an original one. The paper is well structured, the new material as well as LC mixture based on it is thoroughly studied and comprehensively described. The developed LC mixture meets the high standards of the industry. The paper will be useful for those dealing with synthesis and formulation of LC materials. I recommend to publish it, provided that the authors address the following remarks:
(1) Line 26. “heilmeiert” should be corrected to “Heilmeier”. The corresponding reference [3] is probably incorrect, should be changed to a relevant paper by Heilmeier, such as “Heilmeier GH, Zanoni LA. Electro-optical device” US patent 3,499,112 A, 1967” or “Heilmeier GH, Zanoni LA, Barton LA. Dynamic scattering: a new electrooptic effect in certain classes of nematic liquid crystals. Proc IEEE. 1968;56:1162–1171”
Answers:
We have corrected “heilmeiert” to “Heilmeier”, and reference [3] was corrected to “Heilmeier GH, Zanoni LA, Barton LA. Dynamic scattering: a new electrooptic effect in certain classes of nematic liquid crystals. Proc IEEE. 1968;56:1162–1171”.
(2) The comparison of LCD to Cathode Ray Tube seems a bit out-of-date, I suppose that attention to competing of FFS with IPS and other LCD modes instead would better match with the title and would be beneficial for the introduction.
Answers:
We removed the comparison of LCD to Cathode Ray Tube and added the comparison of FFS with other conventional LCD.
(3) The values Cp and VHR should be denoted.
Answers:
Thanks very much for the suggestions. The values Cp and VHR were denoted in Table 1.
(4) Table 3 should contain units, when applicable.
Answers:
Good points. We added the units in Table 3.
(5) Minor errors and typos:
- Lines 27-37. References [4-9] are missing in the text
- Line 91. Typo
- X1, X2, N1, N2 should be denoted when first mentioned
- Formulas (5) and (7) should be changed places
- Line 102 – unclear language: “… it can be reduced by introducing cyclohexane such as formula 6”. Formula 6 does not contain cyclohexane.
Answers:
- The references [4-9] have been noted.
- Line 91 was corrected.
- X1, X2, N1, N2 were denoted.
- The places of formulas (5) and (7) were changed.
- The language in Line 104 was corrected.
Round 2
Reviewer 1 Report
All proposed changes have been made